# *Thiorhodovibrio frisius* and *Trv. litoralis* spp. nov., Two Novel Members from a Clade of Fastidious Purple Sulfur Bacteria That Exhibit Unique Red-Shifted Light-Harvesting Capabilities

**DOI:** 10.3390/microorganisms11102394

**Published:** 2023-09-25

**Authors:** Anika Methner, Steven B. Kuzyk, Jörn Petersen, Sabine Bauer, Henner Brinkmann, Katja Sichau, Gerhard Wanner, Jacqueline Wolf, Meina Neumann-Schaal, Petra Henke, Marcus Tank, Cathrin Spröer, Boyke Bunk, Jörg Overmann

**Affiliations:** 1Leibniz-Institut DSMZ-Deutsche Sammlung von Mikroorganismen und Zellkulturen, Inhoffenstraße 7B, 38124 Braunschweig, Germanysteven.brady.kuzyk@dsmz.de (S.B.K.);; 2Former Institution: Paläomikrobiologie, Institut für Chemie und Biologie des Meeres, Universität Oldenburg, Postfach 2503, 26111 Oldenburg, Germany; 3Bereich Mikrobiologie, Department Biologie I, Ludwig-Maximilians-Universität München, Großhaderner Str. 2-4, 82152 Planegg-Martinsried, Germany

**Keywords:** anaerobic anoxygenic photosynthesis, *Chromatiaceae*, infrared light absorption, microbial mats, photosynthetic gene cluster, purple sulfur bacteria

## Abstract

In the pursuit of cultivating anaerobic anoxygenic phototrophs with unusual absorbance spectra, a purple sulfur bacterium was isolated from the shoreline of Baltrum, a North Sea island of Germany. It was designated strain 970, due to a predominant light harvesting complex (LH) absorption maximum at 963–966 nm, which represents the furthest infrared-shift documented for such complexes containing bacteriochlorophyll *a*. A polyphasic approach to bacterial systematics was performed, comparing genomic, biochemical, and physiological properties. Strain 970 is related to *Thiorhodovibrio winogradskyi* DSM 6702^T^ by 26.5, 81.9, and 98.0% similarity via dDDH, ANI, and 16S rRNA gene comparisons, respectively. The photosynthetic properties of strain 970 were unlike other *Thiorhodovibrio* spp., which contained typical LH absorbing characteristics of 800–870 nm, as well as a newly discovered absorption band at 908 nm. Strain 970 also had a different photosynthetic operon composition. Upon genomic comparisons with the original *Thiorhodovibrio* strains DSM 6702^T^ and strain 06511, the latter was found to be divergent, with 25.3, 79.1, and 97.5% similarity via dDDH, ANI, and 16S rRNA gene homology to *Trv. winogradskyi*, respectively. Strain 06511 (=DSM 116345^T^) is thereby described as *Thiorhodovibrio litoralis* sp. nov., and the unique strain 970 (=DSM 111777^T^) as *Thiorhodovibrio frisius* sp. nov.

## 1. Introduction

Anaerobic anoxygenic phototrophic bacteria occur where light reaches anoxic layers in either stratified pelagic/ freshwater habitats or the surface layers of aquatic sediments. Various species of anoxygenic phototrophs have been found with significant adaptations to different physicochemical conditions and absorbing particular wavelengths and intensities of light [1]. In sediments, optical pathlengths are very short and water has comparatively little effect on the spectral composition. Rather, light attenuation is strongly influenced by the physical matrix, and in sandy sediments the intensity of blue wavelength light is rapidly diminished due to the reflection of quartz grains. In dense microbial mats, cyanobacteria and diatoms often colonize the sediment surface and further absorb light between 400 and 700 nm. As a result, mostly far-red and infrared light is available in the lower, sulfidic sediment layers of benthic microbial mats. Anoxygenic phototrophic bacteria in these deeper areas thus compete strongly for wavelengths between 700 and 1100 nm and have adapted through the evolution of specific photosynthetic pigments and protein complexes, with several species exhibiting markedly different absorption properties [1]. 

The photosynthetic antennae that are capable of absorbing the longest wavelengths contain bacteriochlorophyll *a* (BChl *a*) or BChl *b*, and are commonly found in *Alpha-*, *Beta-*, and *Gammaproteobacteria*. When bound to polypeptide reaction center (RC) and light-harvesting (LH) complexes, BChl absorption is shifted to longer wavelengths and typically peaks at 800, 850, and 880 nm for antenna complexes containing BChl *a*, or around 1020 nm for those binding BChl *b*. While absorption between 900 and 1000 nm could be expected to provide selective advantage for benthic anoxygenic phototrophs [2], very few anoxygenic photosynthetic species have been found capable of exploiting this wavelength range to date. Only ”*Roseospirillum parvum*” [2] and *Thermochromatium tepidum* [3] exhibit unusual long-wavelength absorption maxima of BChl *a*-containing LHI at 909 and 918 nm, respectively, whereas *Rhodospira trueperi* features an unusual blue-shifted in vivo absorption maximum at 986 nm for its BChl *b*-containing antenna complexes [4]. However, no species exploiting the wavelength gap between these peaks (918–986 nm) has been described so far.

In a systematic search for novel types of phototrophic sulfur bacteria with different light absorption properties, highly selective illumination conditions with infrared light filters yielded a purple sulfur bacterium of the family *Chromatiaceae* (*Gammaproteobacteria*), provisionally named strain 970. This unique strain contains BChl *a* like most other anoxygenic phototrophic Proteobacteria, but shows an in vivo absorption maximum at 963 nm, which represents the largest infrared-shift documented for LH complexes [5,6]. 

So far, only the structure and biophysical functions of the photosynthetic apparatus of strain 970 have been studied [5,6,7]. Here, we report on the genomic, biochemical, and physiological properties of this unusual anoxygenic phototrophic bacterium, which is described as the novel species *Thiorhodovibrio frisius* sp. nov.

## 2. Materials and Methods

Strain 970 was isolated from orange-red colored patches found on the sediment surface of a marine ditch along the shoreline of Baltrum, a North Sea island in Northern Germany (Figure 1a). During collection in August 1996, the sediment was covered by 40–50 cm of seawater. Samples were first gathered in tubes, and then the remaining headspace topped with in situ seawater, before being transported back to the laboratory. 

For the cultivation of *Chromatiaceae*, a selective anoxic, marine, hydrogen carbonate-buffered medium (CRs) supplemented with sulfide as an electron donor, was inoculated with sediment subsamples. The basal composition was KH_2_PO_4_, 1.84 mM; NH_4_Cl, 6.36 mM; KCl, 4.56 mM; MgSO_4_·7 H_2_O, 14.2 mM; CaCl_2_·2 H_2_O, 1.7 mM; NaCl, 342.2 mM. After autoclaving, the medium was cooled under nitrogen gas to become anoxic and the following ingredients were added: NaHCO_3_, 60 mM; Na_2_S, 1.25 mM; Trace Elements SL-12, 1 mL·L^−1^ [8]. The pH was then adjusted to pH 7.3. As inoculum, 10% of the culture volume was replaced either directly by the sediment sample, or by 1:10^3^ and 1:10^6^ dilutions of the original sample prepared in CRs medium. The dilution factors were chosen based on cell titer estimates of phototrophic cells in the collected sediment. Briefly, BChl *a* was extracted from sediment samples in acetone and spectrophotometrically quantified (see below), applying an average BChl *a* content of 25 µg·(mg cell carbon)^−1^ [9], a conversion factor of 1.21·10^−13^ g cell carbon·µm^−3^ [10], and an average cell volume of 30 µm^3^ determined by light microscopy (see below) of cells in the samples. 

All enrichments were incubated behind a long-pass infrared edge filter with a 50% transmission cutoff at 900 nm (Göttinger Farbfilter GmbH, Bovenden-Lenglern, Germany). A 40 W tungsten lamp bulb was employed for illumination, resulting in a quantum flux of 25 µmol·m^−2^·s^−1^ behind the infrared filter. Pure cultures were obtained from colored enrichments that contained microscopically visible cells with sulfur globules, using repeated deep agar dilution series to achieve axenic cultures [11]. Purity was routinely checked via denaturing gradient gel electrophoresis of 16S rRNA gene fragments or 16S rRNA gene sequencing, and confirmed by microscopic controls of all cultures after each substrate test (see below). Stock cultures were grown in 50 mL screw-capped bottles containing CRs medium supplemented with 3 mM acetate and at 500 µmol quanta·m^−2^·s^−1^ from a tungsten lamp bulb (40 W). Repeated additions of neutralized sulfide solution increased biomass and cellular yield [12]. Stock cultures were stored short-term at 4 °C in the dark, whereas cell suspensions supplemented with 7% (*v*/*v*) DMSO were cryopreserved in liquid nitrogen for long-term conservation.

Cell morphology and purity of cultures were assessed by phase contrast light microscopy (Leitz DM RBE; Leica, Wetzlar, Germany; and Zeiss Axiolab A.1, Carl Zeiss, Oberkochen, Germany). Samples were prepared on agar-coated microscope slides after adding 2 mL of heated 2% *w*/*v* agar solution on the glass, before drying slides overnight [13].

For transmission and scanning electron microscopy (TEM, SEM), cell pellets were resuspended in CRs medium containing 2.5% glutaraldehyde. After one hour of fixation, cells were washed four times with a buffer containing 32 mM cacodylate and 2% (*w*/*v*) NaCl, at pH 7.2. Washing steps lasted for 10 (5), 20 (25), 40 (35), 75 (30) min (specifications in parentheses are for SEM instead of TEM). Samples were then incubated for 90 (30) min in the same buffer containing 1% (*w*/*v*) osmium tetroxide and further washed in the buffer for 20 min (overnight), followed by a wash in 1:2 diluted buffer for 90 (15) min, and finally rinsed in distilled water twice for 35 min and once for 65 min (three times for 30 min). Dehydration took place using a series of increasing concentrations of acetone comprising 10% (*v*/*v*) for 60 min (15 min), 20% with 1% uranyl acetate for 30 min (without uranyl acetate 15 min), 40% for 45 min (overnight), 60% for 15 min (35 min), 80% for 15 min (15 min), and finally 100% for 15 min, 30 min, overnight, and 65 min (20 min, 50 min, for two days) (specifications in parentheses are for SEM instead of TEM). Samples were viewed in a Hitachi S-4100 (Hitachi, Tokyo, Japan) field emission scanning electron microscope and in a Zeiss EM 912 Omega transmission electron microscope with an integrated zero-loss-mode energy filter (80 kV) (Carl Zeiss, Oberkochen, Germany).

Absorption spectra were recorded with a Lambda 2S UV/VIS spectrophotometer scanning between 325 and 1100 nm (Perkin Elmer, Weiterstadt, Germany). Whole cells from the exponential growth phase were suspended in a 70% glycerol solution for in vivo spectra. Pigments were analyzed from cellular biomass of well-grown cultures or enrichments after concentrating cells by centrifugation (20,400× *g*, 10 min; Beckmann, Munich, Germany) or glass-fiber filtration (type GF/F, nominal pore size 0.7 µM, Whatman, Maidstone, UK). Afterward, acetone (99.5% *v*/*v*) or acetone/methanol (7:2) solvents were added and pigments extracted at 4 °C in the dark overnight, and the supernatant analyzed. BChl *a* concentrations were then quantified using a specific absorption coefficient of 92.3 L·g^−1^·cm^−1^ [14]. Carotenoid composition was determined by filtering extracts (0.2 µm pore size) into small dark-colored glass vials, drying under nitrogen gas flow, resuspending in 500 µL of methanol/acetonitrile (85:15), mixed 10:1 with ammonium acetate (1M), and analyzed by LC-MS [15]. Photosynthetic membranes were partially purified by lysing cells via French Press in TRIS HCl pH 7.8 buffer, discarding pelleted debris (20,000× *g*, 30 min), and concentrating membranes overnight (50,000× *g*, Beckmann UltraCentrifuge, Munich, Germany). The cytoplasmic supernatant was removed, membranes washed in buffer and concentrated overnight (50,000× *g*), prior to being resuspended in new buffer and diluted to 1 mg/mL. Treatment using 0.6% LDAO occurred by gently stirring at room temperature (~22 °C) in the dark for 15 min before loading a sucrose density gradient (0.25, 0.5, 0.75, and 1.0 M, each containing 0.05% LDAO), and separating overnight in an ultracentrifuge (50,000× *g*). Spectra were obtained for all individual bands.

Cellular fatty acids were extracted from wet biomass by following the standard protocol of the Microbial Identification System (MIDI Inc., Newark, DE, USA version 6.1) that included saponification and methylation [16]. Compounds were identified against the TSBA40 peak naming table database. Gas chromatography coupled to mass spectrometry (GC-MS) confirmed the MIDI identification and resolved summed features [17]. The exact position of single double bonds was determined after conversion of the fatty acids to their dimethyl disulfide derivatives [18]. Respiratory quinones were purified [16] and analyzed via HPLC coupled to a diode array detector and a high-resolution mass spectrometer [19].

Intact polar lipids were extracted using a modified Bligh and Dyer method [20]. Briefly, cell biomass was mixed with 0.5 mL methanol: dichlormethane (DCM): 0.3% NaCl (2:1:0.8 *v*/*v*/*v*) and 1 µL C_16:0_-C_18:1_-phosphatidylcholine-d82 (Avanti Polar Lipids), as internal standard. Zirconia/silica glass beads were added (ø 1.0 mm), vortexed briefly, and sonicated for 10 min in an ultrasonic bath. After centrifugation (2 min, 12,000× *g*), the supernatant was collected and the cell pellet was treated a second time by adding 0.75 mL methanol: DCM: 0.3% NaCl (2:1:0.8 *v*/*v*/*v*). The extracts were combined and mixed with 0.3 mL 0.3% NaCl and 0.5 mL DCM via brief vortex. After centrifugation, the lower phase was collected and the upper phase was extracted a second time by adding 0.5 mL DCM. The combined DCM phases were dried under a stream of nitrogen. Dried extracts were resuspended in hexane:isopropanol:water (718:271:10 *v*/*v*/*v*) (150 µL/20 mg biomass) and analyzed by HPLC-MS, as described previously [21].

Growth rates at different temperatures, light intensities, pH values, and salinities were investigated by culturing strains in triplicate using CRs medium supplemented with 3 mM acetate to increase cell yield. All experiments were incubated for 10 days using 5% inoculum in 22 mL screw-cap glass tubes with inserted glass beads for suspending cells after inverting each sample, and optical density (OD) was measured at 650 nm by directly inserting them into a spectrophotometer (Bausch & Lomb, Irvine, CA, USA), unless stated otherwise. Light intensities were adjusted by varying the distance of the culture tubes to a daylight fluorescent tube (Osram Lumilux de luxe, 18W), and measured with a LI 250 Light Meter (LI-COR, Lincoln, NE, USA). Salinity tolerances were obtained by adding different amounts of concentrated salt solution (NaCl, 5.06 M; MgCl_2_·6 H_2_O, 0.2 M) to the conventional medium, or to a freshwater medium devoid of NaCl with minimal MgSO_4_·7 H_2_O (0.5 g·L^−1^). In substrate utilization tests, acetate was omitted during medium preparation. Afterward, 1 of 44 different electron donors, electron acceptors, or carbon substrates was added. Nitrogen fixation was investigated in serum bottles closed with butyl rubber stoppers, filled with CRs medium devoid of NH_4_Cl, but containing a gas phase of nitrogen and carbon dioxide (90:10, *v*/*v*). Here, the regular addition of neutralized sulfide solution was provided by syringe and hypodermic needles preflushed with sterile nitrogen.

The sensitivity of strain 970 to oxygen was tested by pouring a freshly grown anaerobic culture into an open flask covered only with a cotton plug. The unsealed flask was then continuously exposed to air on a rotating shaker to facilitate gas diffusion. Subsamples were regularly inoculated into standard anoxic medium to determine the number of surviving cells of strain 970 during prolonged exposure to oxic conditions, with the last sample taken 48 h after transferring into the ventilated flask.

Initial 16S rRNA gene sequence comparisons had shown strain 970 to be a member of the *Gammaproteobacteria* and phylogenetically closely related to *Thiorhodovibrio winogradskyi*, DSM 6702^T^ [5]. Based on the results of our phylogenomic analysis (see below), we compared the chemotaxonomic and physiological characteristics of strain 970 to those of DSM 6702^T^ and of three other related strains: *Trv.* strain 06511 [8], *Rhabdochromatium marinum*, DSM 5261^T^ [22,23], and *Thiospirillum jenense* DSM 216^T^ [24,25,26].

The complete circular genomes of strain 970, *Trv. winogradskyi* DSM 6702^T^ and strain 06511 were sequenced on a PacBio Sequel *IIe* using high molecular weight DNA prepared via a Qiagen Genomic Tip/100 G kit (Qiagen, Hilden, Germany) according to the manufacturer’s instructions. Long-read SMRTbell™ template libraries were prepared according to the documentation provided (Pacific Biosciences, Menlo Park, CA, USA). Briefly, 10 kb libraries required 1 µg genomic DNA first sheared in g-tubes™ (Covaris, Woburn, MA, USA). DNA was then end-repaired and ligated to barcoded adapters by applying components and reactions according to the SMRTbell Express Template Prep Kit 2.0 (Pacific Biosciences, Menlo Park, CA, USA). Samples were pooled via calculations provided by the Microbial Multiplexing Calculator. Conditions for annealing sequencing primers and binding of polymerase to purified SMRTbell™ templates were assessed with the calculator in SMRT^®^link, and final libraries were sequenced on the Sequel *IIe* with one 15h movie per SMRT cell (Pacific Biosciences, Menlo Park, CA, USA). In total, 1/16 of a SMRT cell was run per isolate, each in three separate runs. For all isolates, long read coverages >500× were obtained (Table 1).

Long-read genome assembly was performed with the *Microbial Assembly* protocol included in SMRTlink version 10 using a target genome size of 5.4 Mbp and coverage of 40×. All isolates yielded a single circular chromosomal contig with sizes of 5.41 to 5.47 Mbp (Table 1). 

Annotations were performed with RAST and adjusted to have *dnaA* as the first forward gene. The accession numbers of the three genomes determined in the present study are CP099568 (*Trv. frisius*, strain 970^T^; DSM 111777^T^), CP099569 (*Trv. winogradskyi*, SSP1^T^; DSM 6702^T^), and CP099570 (*Trv. litoralis*, 06511^T^; DSM 116345^T^).

A phylogenomic classification of the three investigated strains was based on the Type Strain Genome Server (TYGS) [27] and the List of Prokaryotic Strains with Standing in Nomenclature (LPSN) [28] for automated 16S rRNA gene, genome wide, and whole-proteome phylogenetic tree generation. They were additionally compared to known anozygenic phototrophs, including 40 *Gammaproteobacteria* genomes from *Chromatiaceae*, *Ectothiorhodospiraceae*, and *Halieaceae*; 3 *Alphaproteobacteria*; and 3 *Betaproteobacteria*, each available in public databases (Appendix A). Information about the type strains of different species was retrieved from Bac*Dive* [29]. A genome-based phylogenetic tree of 43 Proteobacteria was reconstructed from a concatenated amino acid alignment of 92 housekeeping genes generated from the Up-to-Date-Bacterial Core Gene (UBCG) dataset of the EZBioCloud [27]. Manual refinement of the alignment was performed with the edit-option in the MUST package [30]. The application of G-blocks [31] and calculation of maximum likelihood (ML) trees with RAxML v8.2.10 [32] inferred under a GTR+4Γ model used 100 bootstrap replicates for the final product [33]. Average nucleotide identity (ANI) was determined using ChunLab’s online ANI calculator [34] and digital DNA:DNA hybridization (dDDH) was calculated using formula *d4* within the TYGS server provided by the DSMZ [35].

Sequence comparisons of photosynthesis gene clusters (PGCs) were made between the three newly established closed *Thiorhodovibrio* genomes and two closed reference chromosomes available for *Allochromatium vinosum* DSM 180^T^ and *Congregibacter litoralis* DSM 17192^T^. Due to an inconsistent naming of photosynthesis genes within the public databases, automated genome annotations did not allow a rapid and reliable identification, especially for small photosynthesis genes. Accordingly, we completed comprehensive BLASTP and TBLASTN searches using the well-characterized PGCs in *Dinoroseobacter shibae* DSM 16493^T^ as a reference to consistently number and name all genes [36].

Metabolic potential and associated enzymatic pathways were analyzed by extracting nucleotide or amino acid sequences from each strain’s annotated genome as concatenated fasta files input into the KEGG database for functional enzyme predictions (KOALA annotation) with *Ach. vinosum* DSM 180^T^ as the nearest characterized relative [37]. Metabolic pathways from the three *Thiorhodovibrio* strains were also specifically compared for the presence/absence of genes related to salinity and oxygen tolerance, motility, photosynthesis, sulfur, carbon, and nitrogen fixation or energy production, with further analysis of extracted annotations from each genome using Geneious software (2023.0.2).

## 3. Results and Discussion

### 3.1. Selective Enrichment and Isolation of a Novel Anoxygenic Phototroph

Under highly selective illumination with infrared light of wavelengths >900 nm, anoxygenic phototropic sulfur bacteria developed in CRs medium after two weeks of incubation. An enrichment exhibiting unique in vivo absorption with a maximum around 970 nm was selected for the subsequent isolation attempts. The predominant bacterial cells were motile, harbored highly refractile sulfur globules, and contained BChl *a* as indicated by the characteristic long wavelength absorption peak in acetone extracts at 771 nm [38]. Of note, if these early enrichments were subsequently exposed to a full spectrum of light from tungsten lamps, additional types of purple sulfur bacteria appeared in each culture. This indicated that the bacteria absorbing around 970 nm had an advantage only under highly selective illumination conditions.

Repeated deep agar dilution series yielded an isolate designated strain 970. Pure cultures of this strain exhibited an orange-red color (Figure 1b) similar to the sediment surface layer that had originally been sampled, but divergent from other *Thiorhodovibrio* isolates that are pink-red (Figure 1c). Cells were vibrioid to spirilloid, on average 3.6 µm long and 0.9 µm wide, and accumulated intracellular sulfur globules transiently during exponential growth (Figure 1e,f). While originally motile, subcultures lost their motility prior to cryopreservation, with no flagellation or locomotion detected under any of the growth conditions tested during subsequent characterization.

### 3.2. Phylogenomic Placement and Taxonomy of Strain 970

Initial 16S rRNA gene sequence comparisons [5] had shown that strain 970 was related to *Trv. winogradskyi*, DSM 6702^T^. While an early draft genome sequence of strain 970 was found highly related to *Tsp. jenense* DSM 216^T^ [26], additional comparisons of photosynthetic reaction center protein *pufM* genes suggested an even closer relationship of strain 970 to *Rch. marinum* DSM 5261^T^ [26]. However, such phylogenomic considerations were only provisional, since both *Trv. winogradskyi* strains DSMZ 6702^T^ and strain 06511 lacked sequenced genomes, and strain 970 was only available as an incomplete draft. The assembly of circularized chromosomes presented here (Table 1), thus helped to rectify all previous limitations.

Using our new data, a phylogenomic UBCG analysis of 43 photosynthetic strains from *Gammaproteobacteria*, *Betaproteobacteria*, and *Alphaproteobacteria* (Appendix A) captured the phylogenetic depth and diversity of *Chromatiaceae*, as it comprised genomes of 19 different genera. The concatenated alignment of 92 unique marker proteins with 21,501 variable amino acid positions was used to infer the final phylogenomic tree (Appendix A). The additional *Alpha-* and *Betaproteobacteria* sequences together with those of *Halieaceae* and *Nevskiaceae* from *Gammaproteobacteria* rooted the phylogenomic tree, whereas *Ectothiorhodospiraceae* served as an outgroup of the *Chromatiaceae* subtree (Figure 2). With this approach, strain 970 was clearly placed in *Chromatiaceae,* a family found to be highly distinct within *Gammaproteobacteria*, having a long common branch supported by a 100% bootstrap value. The *Chromatiaceae* were further divided into two recognizable and maximally supported clades ranging from strain 970 to *Thiohalocapsa* sp. ML-1 (Clade A) and from *Ach. vinosum* DSM 180^T^ to *Thioflavicoccus mobilis* 8321 (Clade B) (Figure 2). The separation of all taxa in Clade A including the three *Thiorhodovibrio* strains is supported by bootstrap values of 100%, corresponding to the highest possible confidence level. Within Clade A, the three *Thiorhodovibrio* strains formed a monophyletic subclade (Figure 2). The closest relatives of strain 970 were *Thiorhodovibrio winogradskyi* DSM 6702^T^, *Thiorhodovibrio* strain 06511 (formerly *Trv. winogradskyi*; Overmann et al. 1992), *Rch. marinum* DSM 5261^T^ [22,23,39], and *Tsp. jenense* DSM 216^T^ [26]. These findings were additionally supported by 16S rRNA gene-, whole-genome-, and whole-proteome-based phylogenetic trees generated in the TYGS server (Appendix A).

All three *Thiorhodovibrio* strains contained two *rrn* operons in tandem with duplicate identical 16S rRNA genes within their circularized genomes. Pairwise comparisons of 16S rRNA gene sequences from *Thiorhodovibrio* strains 970 and 06511 with that of the only type strain of this genus, *Trv. winogradskyi* DSM 6702^T^ [8], yielded sequence identity values of 98.0 and 97.5%, respectively, and lower values for comparisons with *Rch. marinum* DSM 5261^T^ and *Tsp. jenense* DSM 216^T^ (Appendix A). These levels were below the conservative species delimitation threshold of 98.7% [40], suggesting that all three *Thiorhodovibrio* strains represent distinct species. This conclusion was validated with low calculated dDDH values, ranging between 25.3% and 34.2% for the three *Thiorhodovibrio* spp. (Appendix A), which was far below the species delimitation threshold of 67–73% [41]. Similarly, ANI values for pairwise comparison of the three strains were below the threshold for species demarcation of 95–96% [42] and ranged between 79.1 and 83.7% (Appendix A). Accordingly, *Thiorhodovibrio* strains 970, 06511, and *Trv. winogradskyi* DSM 6702^T^ must be considered three different species of the genus *Thiorhodovibrio*. Strain 970 was therefore designated the type strain of the new species *Thiorhodovibrio frisius* sp. nov. In the original description published 30 years ago, strain 06511 had been assigned to the species *Trv. winogradskyi* due to the absence of molecular data at that time [8]. However, with our updated approach, strain 06511 is also to be considered as the type strain of a new species for which we suggest the name *Thiorhodovibrio litoralis* sp. nov.

### 3.3. Phenotypic Differentiations and Their Genomic Basis within the Genus Thiorhodovibrio

#### 3.3.1. General Morphological, Physiological, and Chemotaxonomical Characteristics

Similar to the other two *Thiorhodovibrio* strains, cells of strain 970 were vibrioid or short spirilla and formed intracellular sulfur globules (Figure 1e,f,g). Transmission electron microscopy also revealed intracellular vesicular membranes akin to its closest four relatives (Figure 1g, Table 2), while strain 970 was covered by an additional extracellular capsule-like layer (Figure 1h). Photolithoautotrophic growth was observed with sulfide and hydrogen as electron donors as for the other two *Thiorhodovibrio* strains (Table 2). Best growth was achieved at concentrations between 1 and 1.25 mM sulfide, limited bacterial replication occurred at 2 mM, and none was detected at higher sulfide concentrations. All three *Thiorhodovibri*o spp. had enhanced growth when supplied excess elemental sulfur, while thiosulfate was solely used by strain 06511. A very limited number of carbon substrates could support photolithomixotrophic growth. Out of the 44 tested substrates, only acetate, pyruvate, and glucose stimulated strain 970. During cultivation with glucose, cells thickened and formed elongated chains of spirilla. 

The temperature optimum during photosynthesis was 27 to 30 °C, with cultivation possible from 15 to 37 °C. Growth was fastest at high light intensities between 50 and 500 µmol·m^−2^·s^−1^. The optimum pH was 7.3, with growth occurring between pH 6.8 and 8.3. Strain 970 showed a high salinity tolerance, enduring between 1.1 and 5.3% (*w*/*v*) NaCl (corresponding to 1.4 to 7% salts) and favoring 1.5–2.1% (2.0 and 2.8% salt content). The genetic adaptations of phototrophs to osmotic pressure were recently thoroughly reviewed [39], revealing a common production of betaine and ectoine solutes to protect cell osmolarity. Both *Trv.* strain 06511 and *Rch. marinum* were previously found to have glycine-sarcosine methyltransferase (GMT) and dimethylglycine methyltransferase (DMT) that synthesize betaine from glycine, also containing the associated betaine transporters BetT and ProVW1-OpuAC, but each lacking ectoine biosynthesis.

Similar to *Trv.* strain 06511, strain 970 and *Trv. winogradskyi* DSM 6702^T^ contained GMT and DMT providing the capability to synthesize betaine as their major cellular osmotic pressure protectant (Appendix A). Furthermore, all three had *proVW1-opuAC* transporter genes, each lacked ectoine biosynthesis, whereas only strain 06511 encoded the additional BetT/L transporter reported earlier [39]. In comparison, betaine and ectoine biosynthesis and corresponding transport systems were not present in *Tsp. jenense*, as expected for a freshwater phototroph [39]. Regarding sodium efflux and transport mechanisms, every *Thiorhodovibrio* species additionally had various sodium symporters (*DASS, SSS, SNF*) and antiporters (*nhaD, mrpF, nhaK*) (Appendix A), many of which were among *Rch. marinum* and altogether absent in *Tsp. jenense*, further revealing specific adaptations to their saline environments. Uptake of osmolytes constitutes a less energy-consuming way of osmotic adaptation than their biosynthesis. Compatible solutes that leak from living cells or are released by lysing cells are expected to reach high concentrations in dense microbial mats [39]. Besides the biosynthesis of compatible solutes, systems for their uptake represent a key adaptation of bacteria colonizing microbial mats in marine sediments [43]. 

Even though strain 970 is an obligate anaerobic bacterium, it was shown to have a high tolerance toward oxygen, since a shaken open-flask culture had viable cells even after 48 h of exposure to air. Notably, the related *Trv. winogradskyi* DSM 6702^T^ could aerobically respire when intervals of anaerobic/microaerobic conditions were controlled within a chemostat [9]. Therefore, these *Chromatiaceae* have microaerophilic capabilities which are likely an adaptation to the conditions in the top layers of sediment, where the bacteria meet low oxygen concentrations [9]. 

While several enzymes specifically detoxify oxygen radicals, including superoxide dismutases, catalases and peroxidases, terminal oxidases in bacterial respiratory chains may also protect against ROS [44]. All three *Thiorhodovibrio* contained [Fe]-superoxide dismutase *sodB*, where both *Trv. winogradskyi* DSM 6702^T^ and *Trv.* strain 06511 had a pair of homologs in addition to a [Cu-Zn] variant *sodC* (Appendix A). These latter two strains also encoded catalase *katG*, while all three had peroxidases *bsaA*, *cpo*, and *garA*, but DSM 6702^T^ solely lacked *ccpA*. Of note, annotations for *cbb_3_*-type cytochrome c oxidase components were present in each *Thiorhodovibrio* sp., but only strain 970 had the assembly protein in addition to subunit II and all other parts, suggesting its capability to produce a functional enzyme. Moreover, each featured numerous oxidases, such as *cydAB* quinol and heme-based oxidases (Appendix A), likely contributing to their oxygen tolerance observed [44].

By contrast, the genome of oxygen-sensitive *Tsp. jenense* DSM 216^T^ only contained a diheme bacterial cytochrome c peroxidase, BCCP; [Fe]-superoxide dismutase, *sodB*; and quinol oxidase, cydAB; whereas the genes for catalase and peroxidase were absent [26]. The *fixNOQP* type heme/copper cytochrome (*cbb_3_*-type) oxidase found in most other *Chromatiaceae* was also lacking in *Tsp. jenense* DSM 216^T^, which might explain its particular sensitivity towards molecular oxygen [26].

While the cultured representatives of all four related species were motile and flagellated (Table 2), *Trv. frisius* strain 970 had lost its motility quite early during the enrichment and isolation process. Evaluating *Thiorhodovibrio* sp. genetic components, which comprised full sets of *flg* and *fli* gene cassettes for flagellated motility (Appendix A), *Trv. frisius* had two notable differences. KEGG-based PANDA annotation detected neither *fliO* (Thiofri_04660) required for flagellar biosynthesis [45], nor *flgM* (Thiofri_04678), a negative regulator of flagellin synthesis [46], yet both were indeed present in the RAST annotated genome (Appendix A). Upon review, the *fliO* gene of strain 970 had a ~60 bp insertion when compared to other *Thiorhodovibrio*, potentially upsetting its function. Furthermore, while all three had an operon with *flgAMN-fliK-flhB*, *Trv. frisius* contained an additional unique 135 nt open reading frame between *flgM* and *flgN* (Thifri_04679). This new gene may be disruptive due to its proximity to a regulator *flgM* or disturb the transcription of genes downstream in the operon, while the modified *fliO* may also be the cause of *Trv. frisius* loss in motility. Future work is needed to confirm either case.

Earlier ^31^P-NMR measurements had detected phosphatidylglycerol as the dominant phospholipid for *Trv. frisius* strain 970, followed by diphosphatidylglycerol and phosphatidylethanolamine [7]. While the other two *Thiorhodovibrio* spp. also had lysophosphatidylethanolamine and some unusual glycolipids, strain 970 did not (Appendix A). All three contained the respiratory quinones MK-8 and Q-8 (Table 2). The fatty acid profiles were also very similar for all three *Thiorhodovibrio* species, with the exception of myristic acid only detectable in strain 970 whereas *Trv. winogradskyi* DSM 6702^T^ and *Trv. litoralis* strain 06511^T^ both contained 11-methyl C_18:1_ ω7c fatty acid and lauric acid.

#### 3.3.2. Unusual Light Harvesting

The long-wavelength absorption maximum of strain 970 membranes and whole cells was positioned at 963–966 nm, significantly different from the spectra of most closely related *Trv. winogradskyi* DSM 6702^T^ and strain 06511 (Figure 1d). So far, it represents the only anoxygenic phototrophic bacterium to employ BChl *a*-containing photosynthetic LH antenna complexes for the absorption of infrared light at wavelengths above 918 nm. Among the *Chromatiaceae*, red-shifted BChl *a* containing LH have only been observed in *Thermochromatium tepidum* ATCC 43061^T^, which exhibits a long-wavelength absorption maximum at 918 nm [3], whereas *Thiococcus pfennigii* DSM 8320^T^ and *Thioflavicoccus mobilis* DSM 8321^T^ can utilize light at wavelengths above 1000 nm through the use of BChl *b* [47]. Based on previous structural modeling, a unique amino acid replacement (αLys^48^ → αHis^48^) in the α-polypeptide of strain 970 has been suggested to be involved in the strongly red-shifted absorption band of BChl *a* in its LHI [6]. Cryo-EM structure analysis revealed that the C-terminal domains of LHI bind 16 Ca^2+^ ions that are coordinated by amino acid residues which, due to their vicinity and hydrogen-bonds to the BChl *a* molecule, also cause a red-shift of the strain 970 antenna complexes [7].

As indicated by the major absorption peak at 966 nm, strain 970 contains LHI as the only type of light-harvesting complex and lacks LHII. *Trv. winogradskyi* DSM 6702^T^ and strain 06511 were first found to exhibit peaks at 794 and 867 nm with instrumentation limited to measurements up to 900 nm [8]. Upon reanalysis with current equipment, *Trv. winogradskyi* DSM 6702^T^ and strain 06511 were both shown to have an additional shoulder at 908 nm (Figure 1d). Partially purified photosynthetic membranes of strain 06511 (Figure 3a) revealed an LHI at ~900 nm bound to RC (Figure 3b), and an easily disassociated LHII with maxima around 830–850 nm depending on detergent strength (Figure 3c). The LHII shift from 867 to 830 nm when in complex or alone was abnormal, as most LH do not drift as dramatically when purified [48]. This change may be due to specific complex formation of RC-LHI-LHII, potentially involving Ca^2+^ stabilization as documented for strain 970 [7]. The related *Ach. vinosum* contains both LHII and LHI resulting in absorption peaks at 800 and 852 nm and a shoulder at 875 nm, respectively. However, these *Thiorhodovibrio* appear more similar to distantly related *Thermochromatium tepidum*, which has LH peaks at 800, 852, and 914 nm [49], comparable to the 796, 869, and 908 nm found here. Future study of *Thiorhodovibrio* strains DSM 6702^T^ and 06511 may elucidate by which molecular mechanism such a strong red shift occurs for LHII.

Analyzing the *Chromatiaceae* genomes, it became evident that the photosynthesis genes were dispersed over each chromosome, rather than organized within a single contiguous cluster as in the anoxygenic phototrophic members of the *Alpha-* and *Betaproteobacteria* [50]. While all *Thiorhodovibrio* sp. had some *pufBA* genes (encoding the β and α polypeptides LHI) as part of a canonical operon structure with *pufLMC*, additional *pufBA* were found in different regions (Appendix A).

First considering the LHI-related components, an earlier combined iPCR/cloning and sequencing strategy had demonstrated that some photosynthesis genes in the *puf* operon of strain 970 have a typical order where genes *pufBA* were followed by *pufLMC*, encoding the L- and M- subunits of the photosynthetic RC, and RC-bound tetraheme cytochrome, similar to most phototrophic *Alpha-* and *Betaproteobacteria*. However, the *puf* operon in strain 970 was atypical since it contained a second pair of LHI genes downstream, which yielded an overall operon structure *pufB_1_A_1_LMCB_2_A_2_* [6]. Multiple copies of *pufBA* sequences have similarly been reported for other *Chromatiaceae* such as *Ach. vinosum* and *Lamprocystis purpurea* [51]. Our comparative analysis of the photosynthetic gene cluster in related *Gammaproteobacteria* with closed genomes of *Ach. vinosum* DSM 180^T^ and *Cgb. litoralis* DSM 17192^T^ revealed that the *pufB_1_A_1_LMCB_2_A_2_* gene arrangement occurs in both *Chromatiaceae* subgroups A and B (Figure 4). Based on a cryo-EM study at 2.82 Å resolution, 16 LHI complexes form a multimeric ring-like structure surrounding the RC [7]. A notable unique feature among all *Chromatiaceae* genomes, however, is the observation that the entire *puf* cluster is duplicated in strain 970, and both contain separate combinations of *pufBA* (Figure 3). Overall, the closed genome of strain 970 encodes five *pufA* and four *pufB* genes (Appendix A). Interestingly, four (α1–4) and two (β1 and β4) of these polypeptides, respectively, have been detected in strain 970 cells where they occurred in the same LHC I-complex (6 × α1, 1 × α2, 8 × α3, 1 × α4, 3 × β1, 13 × β4; [7]). The various α- and β-polypeptides observed may be related to the multiple copies of *pufBA* genes in strain 970, which could be expressed at very different frequency, perhaps depending on the light intensity, and with their associated mRNA transcripts having individual stabilities [6].

In addition to BChl *a*, the photosynthetic pigments of strain 970 comprise four different carotenoids. Previous analyses had shown that 3,4,3′,4′-tetrahydrospirilloxanthin was the dominant carotenoid of strain 970, followed by two other carotenoids of the unusual spirilloxanthin pathway, namely 3,4-dihydroanhydrorhodovibrin and 3′,4′-dihydrorhodovibrin, with rhodopin detected in small amounts [5,43]. This carotenoid composition clearly differed from that of the four closely related species and indicated that strain 970 employs the unusual spirilloxanthin pathway (Table 2). 3,4,3′,4′-tetrahydrospirilloxanthin occurs in only few phototrophic bacteria, including the BChl *b*-containing phototrophic *Alphaproteobacterium Rhodospira trueperi* [4] and *Gammaproteobacteria Thiococcus pfennigii*, *Thioflavicoccus mobilis,* and *Thioalkalicoccus limnaeus* [47], but has also been detected in small amounts in the BChl *a*-containing *Alphaproteobacterium Rhodoplanes pokkaliisoli* [52]. Spirilloxanthin, in contrast, occurs much more frequently and represents the dominant carotenoid in about half of the species of anoxygenic phototrophic *Alpha-*, *Beta-* and *Gammaproteobacteria*, and is also present in *Trv. winogradskyi* DSM 6702^T^ and strain 06511^T^ (Table 2).

Spirilloxanthin is synthesized from lycopene through the normal spirilloxanthin pathway involving two consecutive sequences of the acyclic carotene C-1,2 hydratase CrtC, the acyclic carotene C-3,4 desaturase CrtD, and the acyclic 1-hydroxycarotenoid methyltransferase CrtF [52]. A low activity or absence of CrtD results in the synthesis of carotenoids with saturated 3,4 and 3′,4′bonds, i.e., 3,4-dihydroanhydrorhodovibrin, 3′,4′-dihydrorhodovibrin, and ultimately 3,4,3′,4′-tetrahydrospirilloxanthin. These latter synthesis steps require only the activity of CrtC and CrtF. Correspondingly, *crtD* mutants of *Rhodospirillum rubrum* and *Thiocapsa roseopersicina* have been shown to produce 3,4,3′,4′-tetrahydrospirilloxanthin [52]. While all genomes had *crtCEF* located upstream from their *bchCXYZ* and *pufB_1_A_1_LMCB_2_A_2_* operons, *Trv. frisius* had an elongated *crtC*, but lacked *crtD* in this location. Instead, *crtD* was found next to a putative 8′-apo-beta-carotenal 15,15′-oxygenase *crt* gene (Thiofri_02479; Appendix A) and hence might be involved in retinal rather than carotenoid synthesis, which would result in the unusual spirilloxanthin pathway becoming active. Another noteworthy alteration of the photosynthesis genes of *Trv. frisius* strain 970 is the species-specific genetic amalgamation of two proteins for carotenoid biosynthesis, the geranylgeranyl pyrophosphate synthetase, CrtE, and the hydroxyneurosporene methyltransferase, CrtF, to a CrtEF fusion protein (Appendix A). This fusion of two genes that are involved at the beginning and the end of the carotenoid synthesis pathway is expected to affect their regulation, which might lead to the accumulation of the intermediates 3,4-dihydroanhydrorhodovibrin and 3′,4′-dihydrorhodovibrin in the unusual spirilloxanthin pathway.

#### 3.3.3. Sulfur and Hydrogen Metabolism

As expected, *Thiorhodovibrio* genomes encoded proteins known to be involved in sulfide oxidation, including sulfide dehydrogenase *fccAB*, and sulfide:quinone oxidoreductase *sqrD/ sqrF* for the oxidation of sulfide to elemental sulfur (Appendix A), where the latter required manual annotation after comparisons to *Thiocapsa roseopersicina* [53]. In *Ach. vinosum* DSMZ 180^T^, *Tsp. jenense* DSM 216^T^, and other *Chromatiaceae,* the genes encoding dissimilatory sulfite reductase and associated enzymes required for oxidation of sulfur to sulfite are found in a gene cluster *dsrABEFHCMKLJOPNRS* [37]. Accordingly, dissimilatory sulfide reductase genes *dsrABEF* were found in the three *Thiorhodovibrio* strains, and all fifteen were present once compared to *Ach. vinosum* DSMZ 180^T^ (Appendix A). Polysulfide reductase *psrA* of the *psrABC* cassette was detected solely in strain 970 [54], peculiarly placed as one of the seven ORFs situated between its duplicate *puf* clusters. Heterodisulfide reductase *hdrABCD*-associated genes *sdhCDAB* were present in all three species [55]. Additionally, the genes involved in the final conversion of sulfite into sulfate during anoxygenic photosynthesis *sat-aprMBA* were found [56], of which *aprM* had to be curated and only KEGG detected *aprBA* (Appendix A).

The genomes of the three *Thiorhodovibrio* strains also contained some genes involved in assimiliatory sulfate reduction, which is commensurate with the low sensitivity toward molecular oxygen. Sulfate is likely transported using CysA in all three *Thiorhodovibrio* species. The gene encoding the assimilatory form of adenylylsulfate reductase *cysH* was found in both *Trv. winogradskyi* and strain 06511, but these genes were notably flanked by transposases (thiowin_04138/04156; thiosp_02721), likely inferring recent acquisition, particularly due to the absence of any other *cys* genes. In comparison, *Tsp. jenense* DSM 216^T^ lacked all genes for assimilatory sulfate reduction (*cysAWTPBIHDN*) [57], including *cysH*, sulfite reductase (*cysI*), and those encoding a sulfate uptake system (*cysAWTP*) being absent [26].

Thiosulfate utilization is mediated by *sox* genes [58]. In *Ach. vinosum*, *sox* genes occur in three separate clusters, *soxYZ*, *soxB*, and *soxXA*, all of which are required for thiosulfate oxidation [37]. While each *Thiorhodovibrio* genome contained *soxYZ*, only *Trv. litoralis* strain 06511 had *soxB* as well as *soxXA* completing the thiosulfate oxidation pathway (Appendix A). This supported the observed physiology since only strain 06511 was observed to grow with thiosulfate (Table 2). The presence of both *soxYZ* and *soeABC* in all three bacteria indicates each could perform sulfite oxidation similar to *Ach. vinosum* [59].

Sulfur globules have also been observed in all three *Thiorhodovibrio* strains (Figure 1e,g). Three hydrophobic proteins, *sgpA*, *sgpB*, and *sgpC*, have been suggested to be associated to sulfur-globule envelopes [60], of which *sgpA* was annotated twice in the KEGG pathways as *sgpA-1* and *sgpA-2* in both *Trv. frisius* and *Trv. winogradskyi* (Appendix A) but not in *Trv. litoralis*. However, Geneious-based alignment analysis showed these genes to be present in the latter species as well, where *sgpA-1* and *sgpA-2* correspond to the canonical *sgpA* and *sgpB*. A lack of *sgpC* could be the reason why the size of the intracellular sulfur globules in *Thiorhodovibrio* spp. is smaller than that in bacteria with all three genes [60].

Genes for several hydrogenases were discovered in the three species, including *hndACE*, *hoxGHKY*, *hupC*, *hyaBC*, *hybCDO*, *hyfB*, and *hypABCDE* explaining their use of molecular hydrogen (Appendix A). *Tsp. jenense* DSM 216^T^ contained the genes *hoxHMFYUFE* that encode a hydrogenase and used hydrogen, whereas *Rch. marinum* DSM 5261^T^ lacked most related genes and correspondingly did not use hydrogen.

#### 3.3.4. Carbon Metabolism

All three *Thiorhodovibrio* species had identical carbon fixation pathways using the complete Calvin–Benson–Bassham cycle including RuBisCo and phosphoribulokinase genes (Appendix A). In addition, they also contained a reductive tricarboxylic acid cycle and its associated enzymes *idh12*, *ppc*, and *por/nifJ* (Appendix A), providing an alternative carbon fixation method. Regarding the tricarboxylic acid cycle, it seems to be incomplete and malate dehydrogenase is lacking, in line with the very limited capability to assimilate organic carbon substrates, similar to *Tsp. jenense* DSM 216^T^. Furthermore, *Thiorhodovibrio* spp. also contained an entire glycolysis/gluconeogenesis pathway that could convert acetate to glucose. This was supported by phenotypic analysis, as both pyruvate and acetate were utilized by all three species (Table 2).

Interestingly, only *Trv. frisius* could metabolize glucose (Table 2), developing atypical thicker cell walls and displaying a thin capsule-like extracellular formation when grown on this monosaccharide (Figure 1h). However, none of the *Thiorhodovibrio* strains contained the recognizable exogenous glucose uptake system *crr*, or any monosaccharide transporters for glucose such as *glcSTUV* or *gtsABCmalK* [61,62]. Upon further review of the KEGG-annotated ABC transporters, however, the *Trv. frisius* genome was unique as it contained genes for a complete capsular polysaccharide ABC-2 transporter system *kpsEMT*. These genes are known to transport polymers of glucose and saccharides for capsule formation [63]. Some ABC transporters have been suggested as bidirectional, capable of importing or exporting small organics such as amino acids [64,65]. The KpsEMT system may thus be involved in glucose uptake and/or the capsule formation of *Trv. frisius*, warranting follow-up analysis.

#### 3.3.5. Nitrogen Metabolism

As with carbon fixation, similar nitrogen-associated pathways existed in each *Thiorhodovibrio* spp. when analyzed using the KEGG online database, and within Geneious. A set of *nifDKH* genes for Fe-Mo dinitrogenase and dinitrogenase reductase was present [66], along with genes for specific regulators including *draGT* [67], conferring nitrogen fixation capability in anoxic environments (Appendix A). Found in Geneious but not in the KEGG annotation, was a complete set of “*Rhodobacter* nitrogen fixation” *rnfABCDEFG* genes, factors important for electron transport to nitrogenase [68]. Each species could additionally perform dissimilatory nitrate reduction to nitrite with *narI*, but not completely to ammonia as they neither had *nirBD* nor *nrfAH*. All *Thiorhodovibrio* spp. lacked the pathways for denitrification, nitrate ammonification, nitrification or annamox capabilities. No assimilatory nitrate reduction was detected in the gene complements, but glutamine and glutamate synthases *gltBD* and *glnA* were found respectively, confirming nitrogen could be fixed into ammonia and then incorporated into amino acids. Similarly, *Tsp. jenense* DSM 216^T^ contained *nifHDKT-nafY* encoding Fe-Mo nitrogenase and the genes for nitrate ammonification [26]. These bacteria thus have a reliance on either an anoxic environment to fix nitrogen, or the presence of ammonia or other fixed nitrogen to grow.

## 4. Conclusions

The whole circularized genomes generated in the present study allowed for a determination of accurate ANI and dDDH values, together with complete operon structures. While current annotation software missed numerous genes for photosynthesis, motility, sulfur-related enzymes, and likely others, our additional manual curation of genes in each *Thiorhodovibrio* strain revealed that observed phenotypes largely corresponded to the gene functions encoded.

On the basis of the characteristics described above, strain 970 = DSM 111777^T^ constitutes a novel bacterial species designated *Thiorhodovibrio frisius* sp. nov. Our results also indicate that strain 06511 = DSM 116345^T^ must be considered a novel species, which is described here as *Thiorhodovibrio litoralis* sp. nov.

## 5. Emended Description of the Genus *Thiorhodovibrio* Overmann et al. 1992

The description is as given by Overmann et al. [8] with the following amendments. Contains carotenoids of the normal or unusual spirilloxanthin pathways, and have BChl *a* bound into LHI that can absorb either ~908 or 963 nm. Some species contain LHII. All species are marine, requiring at least 1.1% NaCl. Hydrogen and excess elemental sulfur can be electron donors for photosynthesis in addition to sulfide. Cannot photoassimilate acetoin, L(+)-arginine, L(+)-ascorbate, benzoate, 2,3-butanediol, butyrate, caproate, caprylate, citrate, crotonate, eth-ylene glycol, ethanol, formate, fructose, fumarate, gluconate, L(+)-glutamate, glycerol, glycollate, lactate, malate, malonate, mannitol, methanol, 2-oxo-glutarate, propanol, succinate, tatrate, valerate, casamino acids, peptone, or yeast extract. In addition, butanol, 1,2-propanediol, 1,2-butanediol, and trimethoxybenzoate do not support growth. Contains Q-8 and MK-8 as respiratory quinones, along with lipids PE, LPE, PG, DPG, and glycolipids. Primary to quaternary fatty acids are C18:1 ω7c, C16:0, C16:1 ω7c, and C18:0. GC content is between 60.2 and 61.4% based on complete circularized genomes of all three current members, with total genome sizes 5.42–5.47 Mbp. Plasmids are lacking.

## 6. Emended Description of the Species *Thiorhodovibrio winogradskyi* Overmann et al. 1992

The description is as given by Overmann et al. [8] with the following amendments. In addition to whole cell absorbance maxima at 796 and 869 nm from LHII, it contains a peak at 908 nm representing a red-shifted LHI. Cannot use thiosulfate as an electron donor. Can photoassimilate acetate, propionate, and pyruvate, but not glucose. Contains minor amounts of LPG in addition to the polar lipids mentioned in the genus description. The minor differentiating cellular fatty acids include quinary and senary 11-methyl C18:1 ω7c and C12:0, respectively. The genome of the type strain comprises 5.48 Mb with 4833 protein coding regions, and 60.6% DNA G+C content (from complete genome sequence).

The type strain is SSP1^T^ (=DSM 6702^T^) isolated from a sample obtained on 3 October 1989 from Mahoney Lake sediment, British Columbia, Canada. The GenBank/EMBL/DDBJ accession numbers for the 16S rRNA gene sequence and genome sequence of strain SSP1^T^ are NR_119251.1 and CP099569, respectively.

## 7. Description of *Thiorhodovibrio litoralis* sp. nov.

*Thiorhodovibrio litoralis* (1i.to.ra’lis. L. adj. *litoralis*, at the beach or coast, referring to the shoreline habitat).

Short spirilloid cells 3.2 ± 0.7 µm long and 0.8 ± 0.1 µm wide. Motile by a monopolar monotrichous flagellum. Cultures are pink-red colored. Optimum growth occurs at 33 °C, pH 7.2, and 2.2–3.2% NaCl. Photosynthetic, with absorbance maxima of whole cells at 796 and 869 nm from LHII-binding BChl *a*, and a peak at 908 nm representing a red-shifted LHI. Contains auxiliary pigments of the normal spirilloxanthin pathway including rhodopin, spirilloxanthin, lycopene, and anhydrorhodovibrin. Can use thiosulfate as an electron donor in addition to sulfide and hydrogen during photosynthesis. Acetate and pyruvate are photoassimilated while propionate and glucose are not. Does not require vitamin B_12_. The minor differentiating cellular fatty acids include quinary and senary 11-methyl C18:1 ω7c and C12:0, respectively. The genome of the type strain comprises 5.47 Mb with 4834 protein coding regions, and the DNA G+C content is 61.4% (determined from complete genome sequence).

The type strain 06511^T^ (=DSM 116345^T^ = KCTC 25737^T^) was isolated in 1986 from a laminated microbial mat on the shoreline of Mellum Island, Germany. The GenBank/EMBL/DDBJ accession numbers for the 16S rRNA gene sequence and genome sequence of strain 06511^T^ are OR492544, and CP099570, respectively.

## 8. Description of *Thiorhodovibrio frisius* sp. nov.

*Thiorhodovibrio frisius* (fri’si.us. L. masc. adj. *frisius*, pertaining to Frisia, a region in the northwest of Germany).

Short spirillum or vibrioid cells 3.6 ± 0.8 µm long and 0.9 ± 0.1 µm wide. The type strain lost motility during isolation. Cells form a thin capsular layer. Cultures are orange-red colored. Optimum growth occurs at 27–30 °C, pH 7.3, and 1.5–2.1% NaCl. Photosynthetic, with absorbance maxima of whole cells at 798 and 963–966 nm from LHI-binding BChl *a*, but lacks LHII. Contains auxiliary pigments of the unusual spirilloxanthin pathway including 3,4,3′,4′-tetrahydrospirilloxanthin, 3,4-dihydroanhydrorhodovibrin, rhodopin, and 3′,4′-dihydrorhodovibrin. Cannot use thiosulfate as an electron donor for photosynthesis. Acetate and pyruvate are photoassimilated while propionate is not. Glucose is additionally photoassimilated. Does not require vitamin B_12_. The minor differentiating cellular fatty acids includes quinary C14:0. The genome of the type strain comprises 5.42 Mb with 4922 protein coding regions, and the DNA G+C content is 60.2% (determined from complete genome sequence).

The type strain is 970^T^ (=DSM 111777^T^ = KCTC 25638^T^) isolated from orange-red colored patches found on the sediment surface of a marine ditch along the shoreline of Baltrum, a North Sea island in Northern Germany. The sample was obtained in August 1996. The GenBank/EMBL/DDBJ accession numbers for the 16S rRNA gene sequence and genome sequence of strain 970^T^ are FJ815159.1 and CP099568, respectively.

## Figures and Tables

**Figure 1 microorganisms-11-02394-f001:**
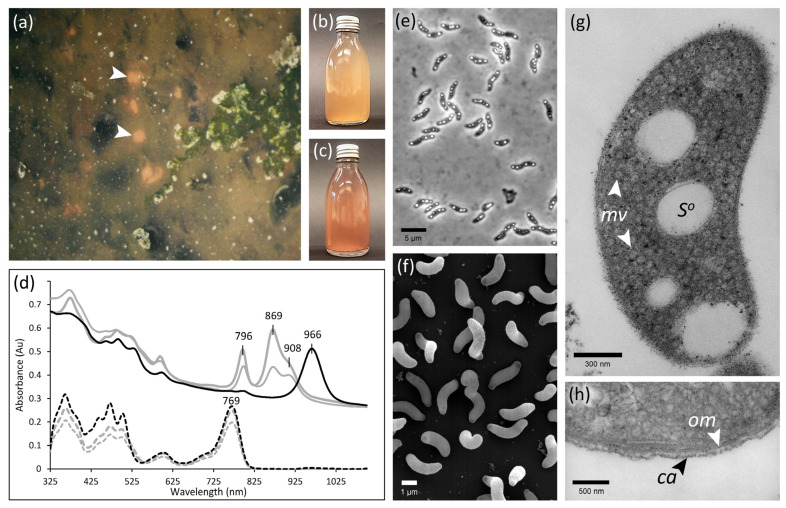
**Natural habitat, culture, and cell morphology of *Thiorhodovibrio frisius* strain 970^T^.** (**a**) Orange-red colored patches (white arrows) at the sediment surface of a marine ditch on the North Sea island of Baltrum (August 1996; courtesy Prof. Dr. Heribert Cypionka, University of Oldenburg). (**b**) Orange-red culture of strain 970. (**c**) Pink-red culture of strain 06511 (*Trv. winogradskyi* DSM 6702^T^ displays the same color). (**d**) Whole cell spectra of strain 970 (black line) compared to strain 06511 (thick gray line) and *Trv. winogradskyi* DSM 6702^T^ (thin gray line), with pigment extracts (7:2 acetone:methanol) in respective colored dashed lines. (**e**) Phase contrast photomicrograph of cells of strain 970 containing highly refractile intracellular sulfur globules. (**f**) SEM micrograph of strain 970. (**g**) Ultrathin section of a whole cell of strain 970 showing void spaces of sulfur globules (S^°^) and vesicular photosynthetic membranes (mv). (**h**) Magnified ultrathin section of a cell showing an extracellular thin capsule-like layer (ca) and outer membrane (om).

**Figure 2 microorganisms-11-02394-f002:**
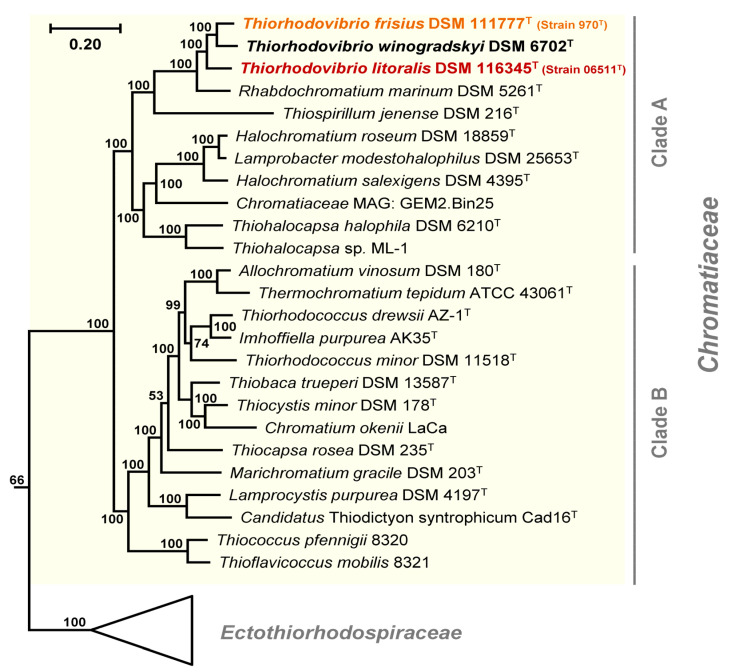
**Phylogenomic species tree based on the analysis of a concatenated alignment of 92 unique marker proteins of 43 *Proteobacteria***. The phylogenetic tree of 37 *Gammaproteobacteria*, 3 *Betaproteobacteria,* and 3 *Alphaproteobacteria* was constructed from 21,501 variable amino acid positions of 92 housekeeping genes. Shown here is the subtree of the *Chromatiaceae* and their sister family *Ectothiorhodospiraceae* serving as an outgroup; the full tree is depicted in Appendix A. Corresponding genome accession numbers are listed in Appendix A. The bar represents relative number of changes per amino acid position.

**Figure 3 microorganisms-11-02394-f003:**
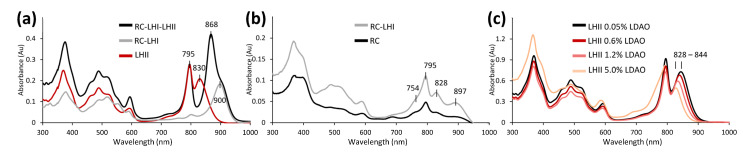
**Photosynthetic membrane spectra of *Thiorhodovibrio* strain 06511.** (**a**) Partially purified RC-LHI-LHII could be degraded to RC-LHI and LHII, (**b**) RC-LHI further separated as RC, or (**c**) LHII with shifting peak depending on detergent treatment.

**Figure 4 microorganisms-11-02394-f004:**
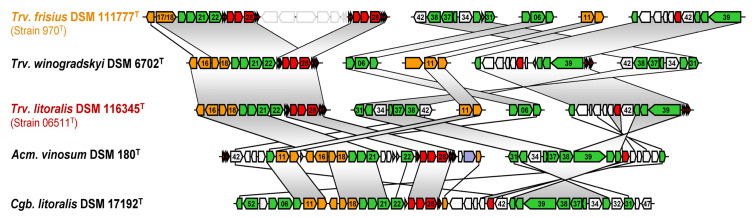
**Comparison of photosynthetic genes.** Three *Thiorhodovibrio* strains, *Ach. vinosum* DSM 180^T^ (*Chromatiaceae, Gammaproteobacteria*), and *Cgb. litoralis* DSM 17192^T^ (*Halieaceae, Gammaproteobacteria*) depicted. Genes are colored according to biological categories: green, bacteriochlorophyll biosynthesis (*bch*); orange, carotenoid biosynthesis (*crt*); red, light-harvesting and photosynthesis reaction center (*puf*); gray, additional conserved genes of the PGC; lilac, Calvin cycle; white, nonconserved genes. Identical photosynthesis gene order indicated by vertical gray areas; single homologous genes are connected with a line. The PGC of *Drb. shibae* DSM 16493^T^ served as reference for the numbering of genes. Gene accession numbers and locus tags are described in Appendix A.

**Table 1 microorganisms-11-02394-t001:** Genomic characteristics of *Thiorhodovibrio* sp. compared to the phylogenomically most closely related *Chromatiaceae*. 1—*Trv. frisius*, DSM 111777^T^; 2—*Trv. winogradskyi*, DSM 6702^T^ [8]; 3—*Trv. litoralis*, DSM 116345^T^ [8]; 4—*Rhabdochromatium marinum*, DSM 5261^T^ [22,23]; 5—*Thiospirillum jenense*, DSM 216^T^ [24,25,26].

Genome Features	1	2	3	4	5
Accession No.	CP099568	CP099569	CP099570	GCA_016583795	GCA_014145225
Size (Mbp)	5.42	5.48	5.47	4.36	3.22
No. of Contigs	1	1	1	214	128
Coverage	911×	1589×	528×	123×	56×
Sequencing platform	PacBio Sequel *IIe*	PacBio Sequel *IIe*	PacBio Sequel *IIe*	Illumina MiSeq	Illumina MiniSeq
GC content (mol %)	60.2	60.6	61.4	59.7	48.7
No. of plasmids	0	0	0	n.d.	n.d.
No. of Genes	5032	5000	4940	3881	2967
No. of CDS	4922	4833	4834	3825	2911
No. of tRNA	51	47	48	46	47
No. of rRNA	6	6	6	6	5

**Table 2 microorganisms-11-02394-t002:** Phenotypic characteristics of *Thiorhodovibrio* compared to the phylogenomically most closely related *Chromatiaceae*.* 1—*Trv. frisius* DSM 111777^T^; 2—*Trv. winogradskyi* DSM 6702^T^ [8]; 3—*Trv. litoralis* DSM 116345^T^ [8]; 4—*Rhabdochromatium marinum* DSM 5261^T^ [22,23]; 5—*Thiospirillum jenense* DSM 216^T^ [24,25,26].

	1	2	3	4	5
Cell morphology	Short spirillum, vibrioid	Vibrio-spirilloid	Short spirillum	Long rods	Curved rods, sigmoid, or spiral
Cell width (±S.D.)	0.9 (±0.1) µm	1.4 (±0.2) µm	0.8 (±0.1) µm	1.5–1.7 µm	2.5–4.0 µm
Cell length (±S.D.)	3.6 (±0.8) µm	3.3 (±0.7) µm	3.2 (±0.7) µm	16–32 µm	30–100 µm
Motility	− (lost)	+	+	+	+
Flagellation	−	Monopolar,	Monopolar,	Bipolar,	Monopolar
		monotrichous	monotrichous	polytrichous	(rarely bipolar),
					polytrichous
** *Carotenoid composition (%)* **					
Lycopene	− ^1^	6.4	14.9	83.3	12
Rhodopin	2.4 ^1^	47.4	55.3	4.8	88
*Carotenoids of the normal spirilloxanthin pathway*					
Anhydrorhodovibrin	−	13.6	10.2	3.5	−
Rhodovibrin	−	4.1	0	8.3	−
Spirilloxanthin	−	28.5	19.6	0	−
*Carotenoids of the unusual spirilloxanthin pathway*					
3,4-Dihydroanhydrorhodovibrin	8.1 ^1^	−	−	−	−
3′,4′-Dihydrorhodovibrin	1.9 ^1^	−	−	−	−
3,4,3′,4′-Tetrahydrospirilloxanthin	87.6 ^1^	−	−	−	−
***Chemotaxonomy*** **^2,3^**					
Quinones (%)	Q8 (84.4), MK8	Q8 (87.4), MK8	(71.0), MK8	n.d.	n.d.
	(15.6)	(12.6)	Q8 (29.0)		
Lipids	PE, LPE, PG,	PE, LPE, PG,	PE, LPE, PG,	n.d.	n.d.
	LPG, DPG,	LPG, DPG,	LPG, DPG,		
	glykolipids	glykolipids	glykolipids		
Fatty acids (%)	18:1ω7c (34.8),	18:1ω7c (46.1),	18:1ω7c (37.2),	n.d.	n.d.
	16:0 (30.7),	16:0 (25.6),	16:0 (28.7),		
	16:1ω7c (25.7),	16:1ω7c (17.8),	16:1ω7c (22.6),		
	18:0 (2.5), 14:0	18:0 (4.3), 11	18:0 (4.5), 11		
	(1.3)	methyl 18:1ω7c	methyl 18:1ω7c		
		(1.7), 12:0 (1.1)	(1.8), 12:0 (1.6)		
** *Physiology* **					
pH optimum	**7.3**	**7.2**	**7.2**	**7.2–7.3**	**7.0**
pH range	6.8–8.3	7.0–7.4	7.0–7.4	n.d.	6.5–7.5
Temperature optimum (°C)	**27–30**	**33**	**33**	**30**	**20–25**
Temperature range (°C)	15–37	14–37	14–37	8.0–35	n.d.
NaCl optimum (% *w*/*v*)	**1.5–2.1**	**2.2–3.2**	**2.2–3.2**	**1.5–5.0**	0
NaCl range (% *w*/*v*)	1.1–5.3	2.2–7.2	2.2–7.2	1.0–6.5	0
Vitamin B_12_ required	−	−	−	−	+
*Electron donors of photosynthetic growth (final concentrations tested)*					
H_2_S (1.25 mM)	+	+	+	+	+
Sulfur (in excess)	+	+	+	+	+
Na_2_S_2_O_3_ (5 mM)	−	−	+	+	n.d.
Hydrogen (head space)	+	+ ^3^	+ ^3^	+	−
*Carbon sources assimilated during phototrophic growth (final concentrations tested)*					
Acetate (5 mM)	+	+	+	+	+
Propionate (1 mM)	−	+	−	+	n.d.
Pyruvate (5 mM)	+	+	+	+	n.d.
Glucose (5 mM)	+ ^4^	−	−	−	n.d.

* All strains stained Gram negative, had vesicular intracellular photosynthetic membranes, and contained BChl *a*. S.D., standard deviation, determined measuring 50 individual cells; +, growth; −, no growth; n.d., not determined. No growth of **1**, **2**, **3**, and **4** was observed on the following substrates (mM concentration): acetoin (10), L(+)-arginine (5), L(+)-ascorbate (5), benzoate (2), 2,3-butanediol (5), butyrate (2.5), caproate (0.5), caprylate (0.2), citrate (2), crotonate (0.2), ethylene glycol (5), ethanol (5), formate (2.5), fructose (5), fumarate (5), gluconate (5), L(+)-glutamate (5), glycerol (5), glycollate (5), lactate (10), malate (5), malonate (5), mannitol (5), methanol (5), 2-oxo-glutarate (5), propanol (5), succinate (10), tatrate (2), valerate (0.5), casamino acids (0.1% *w*/*v*), peptone (0.025% *w*/*v*), or yeast extract (0.005% and 0.1% *w*/*v*). In addition, no growth of **1**, **2**, and **3** was observed on the following substrates (mM concentration): 1,2-butanediol (5), butanol (5), 1,2-propanediol (5), trimethoxybenzoate (2). ^1^ Sourced from [43]; ^2^ Q-8, ubiquinone-8; MK-8, menaquinone-8; PE, phosphatidylethanolamine; LPE, lysophosphatidylethanolamine; PG, phosphatidylglycerol; LPG, lysylphosphatidylglycerol; DPG, diphosphatidylglycerol; glycolipid-A or -B, two different patterns shown (Appendix A); ^3^ determined in the present work; ^4^ forming atypical, thicker cells and elongated chains of spirilla.

## Data Availability

Genome sequences are deposited in NCBI GenBank under the accession numbers CP099568 (*Trv. frisius* DSM 111777^T^), CP099569 (*Trv. winogradskyi* DSM 6702^T^), and CP099570 (*Trv. litoralis* DSM 116345^T^).

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
