# Peer review of "Thiorhodovibrio frisius* and *Trv. litoralis* spp. nov., Two Novel Members from a Clade of Fastidious Purple Sulfur Bacteria That Exhibit Unique Red-Shifted Light-Harvesting Capabilities"

_microorganisms, 2023, doi:10.3390/microorganisms11102394_

Round 1

Reviewer 1 Report

This a well-done manuscript, which gives a detailed comparison of two strains of phototrophic sulfur bacteria that were quite similar to Thiorhodovibrio winogradskyi and which were known for quite some years, but never studied and compared in detail. The phenotypic and genomic information gives insight into important properties of all 3 strains and allows their diagnosis.

Genomes and genomic properties of the two strains were compared to Thiorhodovibrio winogradskyi and related species. Phenotypic as well as genetic information on phylogenetic relations, sulfur metabolism and photosynthesis clearly support the recognition of strains 970 and 06511 as two separate and new species of the genus Thiorhodovibrio. A particular highlight is the shift of the long-wavelength absorption of the bacteriochlorophyll a maximum above 900 nm and as far as 966 nm in Thiorhodovibrio frisius.

Type strains have been assigned, the genome sequences deposited in GenBank, descriptions of the two new species and emended descriptions of the genus Thiorhodovibrio and of Thiorhodovibrio winograskyi were also given.

 It should be noted that for a valid description of the new species a deposition in a second recognized culture collection is required.

Author Response

Response: We appreciate both the Reviewer’s comments and their complements to our manuscript. Our intention was to have a thorough analysis of these new purple sulfur bacterial species, linking their unique and various physiology to genetic determinants, and are happy to have completed a detailed description of these fascinating phototrophs.

Both new species were indeed already deposited to two culture collections, the DSMZ and the KCTC. While one accession number was missing in the submitted manuscript, it has now been added to the revised version as we recently received the deposition certificate and can now list it. 

Reviewer 2 Report

The manuscript (microorganisms-2621891) is well written, and the conclusions are undoubtedly of interest to those that study anoxygenic phototrophs and bacterial taxonomy. The methods are very thoroughly described. The manuscript is a refreshingly complete and scientifically sound work, and it is clear that it was carefully crafted. I have just a few comments/edits to suggest for the authors.

1) Line 38: These phototrophs also occur in stratified freshwater habitats, not just those that are pelagic.

2) Line 85: Metabolisms may be "anaerobic." Growth conditions in culture media lacking O2 would be better described as "anoxic."

3) In Fig. 1d, it looks like only 2 spectra are shown for pigment extracts. I see a black dashed line and a light grey dashed line. Where is the third spectrum for the extracted pigments?

4) In the legend for Fig. 1, panel (f) is described as an "SEM microphotograph." This is simply an "SEM micrograph." Having "photo" in the name suggests light microscopy, which isn't the case here.

5) Line 157: I suggest spelling out "room temperature" and then indicating in parentheses exactly what this temperature was. 22oC? 23oC?

6) Is there no 16S rRNA and/or whole-genome tree included in the ms? I see the concatenated protein tree as Figure 2, but I see no 16S or whole-genome tree in either the paper or the supplementary materials. These would be nice additions to this paper, and I could easily argue having at least one or the other is essential in a species description of this type.

7) Lines 351 and 607-609 mention a capsule present on the surface of cells of strain 970 when grown on glucose, and the legend of Fig. 1 discusses this in reference to panel (h). I see little to no conclusive evidence of a capsule in the TEM shown. Capsules are usually thick enough to be clearly visible in micrographs such that I would expect the capsule to be easily seen even in the cell shown in panel (g), yet what I see looks like the typical "fuzzy" look of LPS in a gram-negative cell wall. Did the authors perform a capsule stain? If so, and it clearly showed the capsule, I suggest adding a picture of this to the ms. At the very least, including a comparative TEM image of a cell grown in medium lacking glucose (and therefore, having no capsule) would help substantiate this.

8) Lines 398: "latter" not "later"

9) In the footnote to Table2, is the "+, (+), -" description of growth based solely on visual inspection, or were optical densities determined for the growth studies? It would be best to indicate an OD range for each symbol if the data are available.

10) Lines 493 and 497: The authors mean "Figure 4" here, correct?

11) Do the authors have any thoughts on how the accumulation of intermediates in the spirilloxanthin pathway might affect growth of the organism?

12) Line 584: "sgpA, sgpB, and sgpC" are names of the genes. If the proteins these genes encode are being referred to here, these should be "SgpA, SgpB, and SgpC."

Author Response

Response: We thank Reviewer #2 for their time and consideration of our manuscript, and are delighted that they find our work to be well written, interesting, and carefully crafted. It was our intention to present an in-depth description of these new species, and to thoroughly compare physiology to genomic content.  All specific comments have been considered, with individual responses listed below:

1) Line 38: These phototrophs also occur in stratified freshwater habitats, not just those that are pelagic.

Response: Indeed. “Freshwater” is now included in the sentence.

2) Line 85: Metabolisms may be "anaerobic." Growth conditions in culture media lacking O2 would be better described as "anoxic."

Response: Agreed. “anaerobic” has been changed to “anoxic”.

3) In Fig. 1d, it looks like only 2 spectra are shown for pigment extracts. I see a black dashed line and a light grey dashed line. Where is the third spectrum for the extracted pigments?

Response: The extracts of strains 06511 and SSP1 were identical, so we had originally chosen to avoid such overlap. However, we have re added it to the updated figure.

4) In the legend for Fig. 1, panel (f) is described as an "SEM microphotograph." This is simply an "SEM micrograph." Having "photo" in the name suggests light microscopy, which isn't the case here.

Response: Corrected.

5) Line 157: I suggest spelling out "room temperature" and then indicating in parentheses exactly what this temperature was. 22oC? 23oC?

Response: Now spelled out, and ~22oC is listed.

6) Is there no 16S rRNA and/or whole-genome tree included in the ms? I see the concatenated protein tree as Figure 2, but I see no 16S or whole-genome tree in either the paper or the supplementary materials. These would be nice additions to this paper, and I could easily argue having at least one or the other is essential in a species description of this type.

Response: Since 16S rRNA gene similarities, DDH, ANI, and a MLSA phylogenetic tree of 92 house keeping genes was provided, additionally phylogenetic trees were not considered necessary. However, we now have included 16S rRNA gene, genome and proteome trees in supplemental material.

7) Lines 351 and 607-609 mention a capsule present on the surface of cells of strain 970 when grown on glucose, and the legend of Fig. 1 discusses this in reference to panel (h). I see little to no conclusive evidence of a capsule in the TEM shown. Capsules are usually thick enough to be clearly visible in micrographs such that I would expect the capsule to be easily seen even in the cell shown in panel (g), yet what I see looks like the typical "fuzzy" look of LPS in a gram-negative cell wall. Did the authors perform a capsule stain? If so, and it clearly showed the capsule, I suggest adding a picture of this to the ms. At the very least, including a comparative TEM image of a cell grown in medium lacking glucose (and therefore, having no capsule) would help substantiate this.

Response: While to our opinion the capsule presented in Fig1F is clearly defined, we agree that it is not a canonical, thick and prominent capsule. Since thin outer layers are difficult to stain efficiently to visually confirm their constituents, we have decided to rephrase the statements to instead refer to this as an “additional extracellular layer”, and deduce that it is a “extracellular thin capsule-like carbohydrate layer” based on the genomic evidence.

8) Lines 398: "latter" not "later"

Response: Corrected.

9) In the footnote to Table2, is the "+, (+), -" description of growth based solely on visual inspection, or were optical densities determined for the growth studies? It would be best to indicate an OD range for each symbol if the data are available.

Response: Arbitrary cut of for the single test (Hydrogen utilization) has been removed. All showed increased biomass, so all are now simply reported as positive “+”, or negative “-“.

10) Lines 493 and 497: The authors mean "Figure 4" here, correct?

Response: Corrected.

11) Do the authors have any thoughts on how the accumulation of intermediates in the spirilloxanthin pathway might affect growth of the organism?

Response: The accumulation of different carotenoids, or even the production of alternative light harvesting complexes may be a result of environmental factors such as light intensity/ spectra or redox changes. We predict that certain pigments and complexes may absorb the provided light better than other combinations, thus enhance growth during specific conditions. As carotenoids act as antioxidants, these can be overproduced in the cell, unbound to RC-LH, and help scavenge oxygen radicals, thus helping the bacteria survive otherwise toxic oxic conditions.

While various LH were detected among the RC-LH structure via cryoEM of Trv. frisius in the preceding paper to this work, no follow-up on the production of different LH or carotenoids has yet been performed. It would be interesting to determine what conditions stimulate specific pigments and complexes, but this kind of analysis would be a whole project into itself, and is out of the scope of this current taxonomic manuscript.

12) Line 584: "sgpA, sgpB, and sgpC" are names of the genes. If the proteins these genes encode are being referred to here, these should be "SgpA, SgpB, and SgpC."

Response: Italics removed.